# Applications of Flow Cytometry in Drug Discovery and Translational Research

**DOI:** 10.3390/ijms25073851

**Published:** 2024-03-29

**Authors:** Sumana Ullas, Charles Sinclair

**Affiliations:** Flagship Pioneering, 140 First Street, Cambridge, MA 02141, USA; sullas@flagshippioneering.com

**Keywords:** flow cytometry, drug discovery, cell biology, pharmacology, oncology, immunology, immuno-oncology, vaccines, translational research

## Abstract

Flow cytometry is a mainstay technique in cell biology research, where it is used for phenotypic analysis of mixed cell populations. Quantitative approaches have unlocked a deeper value of flow cytometry in drug discovery research. As the number of drug modalities and druggable mechanisms increases, there is an increasing drive to identify meaningful biomarkers, evaluate the relationship between pharmacokinetics and pharmacodynamics (PK/PD), and translate these insights into the evaluation of patients enrolled in early clinical trials. In this review, we discuss emerging roles for flow cytometry in the translational setting that supports the transition and evaluation of novel compounds in the clinic.

## 1. Introduction

Flow cytometry is a foundational technique for characterizing and understanding cell populations down to the resolution of a single cell. It allows the ability to discriminate cell size, granularity, and marker expression based on fluorescent and light-scatter properties, enabling deep characterization of cellular and molecular phenotypes. Continuous advancements in hardware and fluorescent dye development have made flow cytometry increasingly quantitative. As a result, flow cytometry plays a growing role in drug discovery and development across multiple disease areas.

The most widely used version of the technique, termed multicolor flow cytometry, is capable of discerning >15 parameters [1]. In a typical example of this method, single-cell suspensions are stained with fluorophore-conjugated antibodies, and the sample is suspended in a buffered salt solution. The sample is injected into the flow cytometer and hydrodynamically focused so that single cells flow past multiple lasers and detectors with precise timing. The lasers excite the fluorescently conjugated antibodies, and the emitted light is collected in a series of detectors (Figure 1). While representing an incredibly powerful technique, there are a number of intrinsic limitations that have driven extensive technology development in recent years. Spectral overlap of emitted light from fluorescent antibodies requires mathematical compensation to subtract the measurement of photons overspilling into multiple detectors. This limitation has driven the technological development of additional lasers and detectors that reduce fluorescence overlap and improved fluorophores with more narrow emission spectra. More recently, mass cytometry was developed to measure conjugated antibodies by the mass of attached metal isotopes [2]. Spectral flow cytometry similarly reduces the impact of spectral overlap and loss of signal from compensation by utilizing spectral unmixing to increase the resolution and sensitivity of the technique [3]. Imaging flow cytometry was developed to enable subcellular visualization of the florescent antibody or dye localization and cell morphology, providing additional insights into signaling events at a single cell level [4]. As a further extension of this, imaging mass cytometry now enables the simultaneous integration of spatial information with quantitative interrogation of markers on single cells [5]. Finally, acoustic flow cytometry has emerged to address instability in hydrodynamically focused samples and greatly increases the throughput of sample acquisition [6]. The evolution of flow cytometry technology has enabled more practical applications to address questions in drug discovery.

Drug discovery pipelines follow a broadly similar roadmap of activities and questions centered around identifying and validating a novel therapeutic (Figure 2). Once a drug target has been identified, initial ‘hit’ molecules are identified through high-throughput screening approaches. Typically, the aim is to find molecules that serve as a starting point for further lead optimization of potency and selectivity or to improve pharmaceutic properties related to absorption, distribution, metabolism, and/or excretion. Alongside this, the molecules must be well characterized in both in vitro and in vivo systems to understand how drug exposure over time (pharmacokinetics, PK) influences pharmacodynamics (PD) and preclinical efficacy. The PK/PD/efficacy relationship can be quantitatively modeled to predict therapeutic doses, enabling clinical trial design. Flow cytometry plays a role throughout the drug discovery process and is also increasingly applied to measure biomarker modulation in early clinical trials, providing insights on the forward progression of a molecule and reverse translation of patient data, enabling the next wave of discoveries.

## 2. Hit Identification and Lead Optimization

Flow cytometry has been deployed early in the drug discovery process to support hit identification. Phenotypic screening is a hit-finding approach where compounds are identified based on their ability to change cellular phenotypes relevant to a disease. Tuijnenburg et al. used flow cytometry to identify compounds that reduced B cell activation and plasmablast formation without affecting cellular viability as potential therapeutics for auto-immune disease. By focusing their characterization on a small collection of kinase inhibitors with known selectivity profiles, the authors identified several inhibitors of the mechanistic target of the rapamycin (MTOR) pathway that could prevent terminal differentiation of plasmablasts [7]. Hu et al. took a broader approach, utilizing high-content imaging-based screening (HCS) to identify initial hits from a library of 4126 compounds that modulated activated macrophage morphology. Flow cytometric measurement of specific surface markers was employed for selected hits, identifying modulators of macrophage M1/M2 polarization [8]. Ding et al. leveraged a high-throughput flow cytometry-based method to directly screen a library of 4213 structurally diverse compounds for modulators of the immunosuppressive transcription factor FOXP3 and its downstream target CTLA4. This screen enabled concurrent identification of FOXP3 stabilizers and destabilizers, which may have relevance to promote immunosuppression for the treatment of inflammatory disorders or to potentiate the immune system as part of an immuno-oncology therapy [9]. Outside of immunology and oncology therapeutic areas, Buranda et al. used flow cytometry to screen for compounds that inhibited the binding of fluorescently labeled hantavirus [10]. Schardt et al. leveraged fluorescence-activated cell sorting (FACS) to purify antigen-specific memory B cells from mice immunized with SARS-CoV-2 antigen to identify clones producing potentially useful neutralizing antibodies [11]. Collectively, flow cytometry-based screens are useful when a target-agnostic or functional screening strategy is preferred and uniquely enables screening in mixed populations of primary cells. Nevertheless, throughput is still somewhat restricted versus other cellular screening techniques such as HCS or homogeneous time-resolved fluorescence (HTRF), which currently limits the application of flow cytometry to screening smaller chemical or protein libraries. Further improvements in automation, multiplexing, and precision sampling/washing will enable greater application at the hit-identification phase of drug discovery [12].

Following compound hit-identification, chemical or biological matter is evaluated to down-select molecular starting points and to guide empirical optimization in a process termed lead optimization (Figure 2). At this stage, drug discovery teams seek to improve key pharmaceutical properties such as potency, safety, PK, and molecular functionality (Table 1). Typically, 10 s–100 s of molecules are profiled iteratively, so medium-throughput assays are suitable. Zhou et al. used flow cytometry to measure binding affinities of single-chain fv (scFV) antibody fragments and bivalent antibodies recognizing the same epitope of EGFR. The assay was sufficiently sensitive to differentiate and rank-order 9 clones that had been affinity matured using yeast display. Data also supported a direct link between affinity, avidity, and downstream functional potency to block EGFR signaling, which guides key parameters for optimizing a potent EGFR therapeutic [13]. Similarly, functional B cell activation assays have been employed in the discovery and optimization of IL-4 receptor alpha (IL-4Rα) antibodies, which link therapeutic potency and activity to a directly relevant metric for Th2 polarized inflammatory disease [14]. Flow cytometric potency assays extend beyond biologics; Wang et al. measured CD69 and CD25 activation markers on activated primary T cells, showing a dose-dependent increase after treatment with HPK1 small molecule kinase inhibitors [15]. Flow cytometry is highly enabling for profiling compounds where reliance on cell lines can be misleading. Antisense oligonucleotides (ASOs) are transported into cells via endocytic pathways, which differ across different cell types and cell lines, leading to highly variable efficacy [16]. Revenko et al. used a flow cytometric assay to triage ~100 ASOs designed to target the lineage-restricted immunosuppressive transcription factor FOXP3. The ability to directly measure protein expression and knockdown in primary cells enabled the nomination of the clinical development candidate AZD8701, which was explored in early-phase clinical trials for the treatment of solid tumors (NCT04504669) [17].

Drug safety is a composite of primary interactions against a desired target and selectivity over off-target pharmacology that may negatively impact host tissues. Highly selective compounds maximize the dosing window where clinical efficacy is observed without concurrent toxicity, referred to as the therapeutic index [18]. Flow cytometry assays have been applied to characterize and mitigate the safety of emerging therapeutics. McDermott et al. utilized flow cytometry in the selection of monoclonal antibodies (mAbs) that bound to the tumor-expressed claudin-6 (CLDN6) with minimal binding to related family members CLDN3 and CLDN9 that are widely expressed in healthy tissues [19,20,21]. Protein kinases are important cell signaling molecules, with inhibitors estimated to represent 25–33% of ongoing drug development efforts in the United States and worldwide [22]. Due to the homologous nature of the kinase active site, selectivity panels have been developed that measure the binding of compounds across the kinome [23,24]; however, this approach does not inform on the functional consequences of binding. Nilsson et al. assessed the selectivity of Janus kinase 1 (JAK1) inhibitors with a phospho-flow cytometry assay to measure inhibition of the primary target (JAK1-dependent phosphorylation of STAT6) and off-target activity (JAK2/TYK2-dependent activation of STAT4) in peripheral blood mononuclear cells (PBMCs). These assays guided the optimization of JAK1 selective inhibitors exhibiting 20–80-fold selectivity windows that were nominated as clinical candidates AZD0449 and AZD4604 [25]. Non-specific biophysical interactions between therapeutics and unrelated host biomolecules can also impart a safety risk. Lower levels of poly-specificity are correlated with improved antibody developability [26]. Makowski et al. recently developed a flow-cytometry based poly-specificity particle (PSP) assay that had higher sensitivity to detect non-specific binding events over conventionally used ELISA methods [27]. The approach described was technically complex, but further simplifications could widely impact drug discovery pipelines. Collectively, the ongoing emergence of highly sensitive, quantitative, and reproducible flow cytometry assays is improving the ability to develop safer therapeutics.

Therapeutic PK is influenced by the physicochemical properties of a molecule, in addition to interactions with the cellular and molecular systems of the host. Flow cytometry assays are particularly relevant to elucidate cellular and molecular factors influencing PK and target tissue exposure. For example, several therapeutic modalities, including mAbs, antisense oligonucleotides (ASOs), or other biologics, may trigger the host immune system, potentially leading to the development of anti-drug antibodies (ADAs). ADAs can impact PK through the neutralization of active therapeutic moieties or increasing drug clearance. Kirkland et al. developed Hu-mAb 2–5, a humanized mouse mAb targeting Shiga toxin (Stx), as a potential treatment for Stx-producing *Escherichia coli* (STEC) infections. Flow cytometric evaluation of IFNγ and TNFα cytokines was leveraged to monitor human immune cell activation potential after repeat exposure in humanized mice [28]. Half-life extension of antibody therapeutics can be achieved by introducing point mutations into the Fc region of the antibody backbone. This increases affinity to the FcRn receptor in the kidney to promote molecular recycling back into circulation [29] but may also inadvertently introduce immunogenic sequences [30]. Ko et al. leveraged flow cytometry as part of an integrated immunogenicity evaluation of a half-life extended version of the anti-TNFα mAb Rituximab. After performing an in silico sequence-based assessment, flow cytometry measurements of T cell proliferation added further empirical support that molecules had a low immunogenicity risk [31]. Modifications to antibodies or proteins have the potential to disrupt primary interactions with the intended therapeutic target. Mandrup et al. developed a bispecific T cell engager platform with ‘tunable’ PK through the introduction of an albumin sequence that promotes recycling into circulation by the kidney. Flow cytometry was used to confirm that both target engagement and efficacy were maintained across albumin sequences with varying half-lives [32]. As these studies demonstrate, flow cytometry is often integrated as part of a broader systems-level evaluation to predict, optimize, and de-risk PK ahead of clinical testing.

Finally, flow cytometry can be used to evaluate the molecular functionality of therapeutics. Antibody and antibody–drug conjugate (ADC) therapies are dependent on antibody-mediated binding, internalization, and trafficking to intracellular compartments and payload release. Advances in pH-sensitive dyes are enabling quantitative measurements of internalization kinetics [33]. Parameswaran et al. used flow cytometry to support the discovery of CD6-targeted ADCs, identifying molecules with a 93.4% internalization efficiency [34]. The application of flow cytometry to measure internalization represents a complementary approach over more commonly used confocal microscopy techniques. Yang et al. confirmed the internalization of a glycosphingolipid targeting ADC OBI-999 in tumor cells using microscopy before switching to a flow cytometry assay to quantify the percentage of OBI-999 internalization over multiple time points [35]. Thus, whilst microscopy allows for more definitive measurements of trafficking to appropriate intracellular compartments, flow cytometry enables more accurate quantitation and higher throughput and quantitation of internalization in a single assay [36,37]. Following internalization, another key property required for ADC efficacy is payload release. Payload release can be related to the antibody target and is additionally dependent on both the linker chemistry and conjugation method [38]. Kopp et al. evaluated the effect of linker stability on the efficacy of a Trop-2 targeting ADC bearing a cytotoxic Topoisomerase I payload, measured by quantitating a specific biomarker of double-stranded DNA breaks phospho-histone H2A.X [39]. In addition to target cell efficacy, cytotoxic payload release of an ADC can also confer a bystander-killing effect through the diffusion of payload molecules to surrounding adjacent cells and tissues [40]. The ability to discriminate populations of cells with a single cell resolution makes flow cytometry a method of choice to explore bystander killing, which may be a favorable therapeutic characteristic if the target antigen is only expressed on a subset of target cells. Bystander killing of HER2 targeting ADCs could be observed in co-cultures of HER2+ and HER2− cells by flow cytometry [41]. An extension of this approach could be used to explore bystander killing efficiencies with different ratios of target-expressing to non-target-expressing cells, albeit this parameter has not yet been studied in detail during ADC development. While these case studies demonstrate the flexibility of flow cytometry to evaluate the multimodal molecular functionality of ADCs, they extend to other modalities that functionally modulate signaling, change target expression through degradation or stabilization, promote cellular adhesion, and beyond. 

Collectively, these examples highlight a diverse application of flow cytometry to impact drug discovery programs, and selected examples are summarized in (Table 2). As the complexity of therapeutic modalities extends beyond small molecules and antibodies, we anticipate flow cytometry to become an increasingly relevant technique during the lead optimization phase of drug discovery.

## 3. Translational Research Informing the Path to the Clinic

It is critical to understand the mechanisms of emerging drug targets and drug candidates in a preclinical setting in order to justify the risk/benefit for patients and support the investment of time, money, and resources spent on future clinical trials. Such evaluation provides a deeper insight into disease and patient selection strategies, differentiation from competitor molecules or standard of care, and can also support a rationale for therapeutic combinations. Flow cytometry has been broadly applied to address meaningful translational questions, unlocking the potential for novel compounds to benefit a greater number of patients.

Evaluation of signaling events and biomarker expression with single-cell resolution has been transformational for the field of immuno-oncology, representing one of the best examples of the impact of flow cytometry to transform the understanding of disease biology. The emergence and approval of antibody therapies targeting CTLA4 and PD-L1 in cancer provided a clinical rationale for harnessing the host immune system with therapeutics to respond against tumors [42,43]. However, the understanding of preclinical immuno-competent syngeneic tumor models and their translational relevance was only recently characterized in detail. In a seminal paper published by a group from Medimmune, flow cytometry and transcriptomics approaches were used to systematically characterize six widely used syngeneic tumor models, revealing a striking difference in the composition of infiltrating immune cells. Models could be categorized as immune-infiltrated or immune-barren by quantifying cells expressing the hematopoietic lineage marker CD45 and revealed differences in immune composition measured by differences in the ratio of CD4+ T cells, CD8+ T cells, and myeloid cells [44]. Additional surface markers can further distinguish cytotoxic immune populations and immune-suppressing cell populations, which exist in homeostatic balance (Figure 3). This differential immune composition is also reflective of human tumors [45], providing insights into their translational relevance. Follow-up studies explored three of the most widely used models, CT26, MC38, and 4T1, further revealing the kinetics of immune recruitment over time, enabling optimal PD timepoint selection, and providing further insight into cellular cross-talk during an immune response to a tumor [46].

The deeper characterization of preclinical immuno-oncology models enables a rational selection of appropriate systems that best recapitulate the microenvironments or pharmacology of interest, enabling a better link between PK/PD and efficacy. Numerous examples now exist where flow cytometric characterization of immunological PD has been central to understanding the mechanism of action of immuno-oncology drug candidates or uncovered previously unknown mechanisms of therapeutic agents. Leyland et al. leveraged flow cytometry in the development of second-generation checkpoint agonists targeting the GITR axis, showing that therapeutic administration to mice bearing CT26 tumors resulted in increased proliferation of CD4+ and CD8+ T cells and reduction of FOXP3+ immuno-inhibitory regulatory T cells [47]. Borodovsky et al. used flow cytometry to elucidate a new mechanism for a small molecule targeting the adenosine 2A receptor (A_2A_R), which enhanced antigen presentation by CD103+ dendritic cells and complemented the direct T cell-mediated effects of immune checkpoint blockade [48]. Our group leveraged the understanding of flow cytometric biomarkers in the tumor microenvironment to better understand the immunological cross-talk of small molecules previously considered to work through tumor-intrinsic mechanisms. Vistusertib targets MTOR kinase and was developed for the treatment of ER+ breast cancer based on its direct effects on tumor cells. MTOR is well established to have immunomodulatory activities [49], and we showed an enhanced anti-tumor efficacy and increased inflammatory profile when vistusertib was combined with immune checkpoint inhibitors targeting CTLA4 or PD-L1. Mechanistically, the combination was associated with increased T effector cell frequencies in vivo, correlating with the direct effects of vistusertib to enhance the viability of weakly activated T cells in vitro, as measured by a flow cytometric proliferation/viability assay [50]. Similar findings were confirmed with inhibitors of the upstream kinase phosphatidyl-inositol-3-kinase (PI3K), supporting the relevance of PI3K/MTOR in an immuno-oncology setting [51]. Moreover, flow cytometry has played a key role in uncovering additional unexpected immune-modulatory functions of several other cancer-targeting drugs, highlighting an important interplay between the immune system and tumor cell biology [52]. Finally, multiparametric characterization with flow cytometry can be used to support a rationale for therapeutic combinations. Wichroski et al. recently reported the discovery of potent small molecules targeting DGKα/ζ, integrating flow cytometry assays to reveal enhancement of TCR signaling under conditions where antigen presentation was rate-limiting [53]. Downregulation of MHC-I and reduced neoantigen presentation is a hallmark of PD-1 or PD-L1 immune checkpoint resistance [54], providing a rationale to combine DGKα/ζ with immune checkpoint blockade in the clinic. Beyond the characterization of existing therapeutic models, flow cytometry has also been used to guide the development of additional models that capture a greater variety of translational pharmacology. The Aryl hydrocarbon receptor emerged as a potential mechanism of immune checkpoint resistance based on clinical datasets [55], but activation of the pathway was not well captured in available preclinical models. Campesato developed a tumor model overexpressing the upstream pathway activator Indoleamine-pyrrole 2,3-dioxygenase (IDO1), showing a direct inhibitory effect on immune checkpoint efficacy, which was directly associated with the detection of CD206+/MHC-II+ co-expressing suppressive myeloid cells into the TME [55]. These IDO1 overexpressing tumors were ultimately leveraged to explore the pharmacology of AHR inhibitors, enabling progression into clinical trials [56]. A novel application of flow cytometry enabled the development of a transplantable, syngeneic peripheral T cell lymphoma (PTCL) model. In this model, flow cytometry was employed to characterize 17 T cell receptor Vβ chains that normally exhibit poly-clonality within healthy T cell populations to promote diversity in their ability to recognize foreign antigens. Transplantation of PTCL led to a progressive clonal outgrowth of a monoclonal population of Vβ8+ PTCL population. In addition, PTCL cells harbored the congenic marker CD45.2, which could be tracked in CD45.1+ congenic hosts. Thus, a combination of Vβ8 and/or CD45.2 represented surrogate efficacy biomarkers, enabling longitudinal tracking of transplanted populations [57]. This new model supported the testing of therapeutics, where it was revealed that PTCLs harbored a sensitivity to the ATR inhibitor AZD6738. Treatment with AZD6738 suppressed the outgrowth of Vβ8+ cells and enhanced the survival of mice, providing a preclinical rationale for further investigation of ATR inhibitors in PTCL patients [58]. We envisage that similar approaches could be applied to other immunocompetent hematopoietic cancer models to better understand the cross-talk between cancer and the host immune system. Whilst these examples focus on studies using flow cytometry to advance the understanding of immuno-oncology therapeutics, these approaches are highly applicable to answer similar mechanistic questions relevant to other therapeutic disease areas.

The ultimate objective of translational research in drug discovery is to bridge the understanding of a therapeutic between preclinical models and patients, which is indicative of future clinical success [59]. Whilst this concept seems simple, experimental measurement is challenging, requiring parallel cross-species studies and careful design of interpretable clinical endpoints. Baumgartner et al. leveraged flow cytometry to evaluate a candidate small molecule drug targeting PTPN1/2, showing that treatment enhanced the expression of the early activation marker CD69 on stimulated mouse splenic T cells in vitro. This result was next recapitulated in a human whole blood assay, supporting future use of this assay in the clinic [60]. Casey et al. applied flow cytometry-based immune cell profiling to characterize the PD changes in clinical samples from lupus patients treated with Anifrolumab, an antibody that blocks signaling through the interferon α receptor I (IFNAR1). Normalization of inflammatory populations was particularly pronounced in interferon-γ signature high populations, supporting the further evaluation of efficacy in stratified patients [61]. Flow cytometry reagents have been historically biased toward model organisms such as mice, which has limited the application when higher-species disease models are more translationally relevant. There has been a concerted effort by the Nonhuman Primate Reagent Resource to catalog cross-reactivity antibody clones across primate species, available at https://nhpreagents.org/reactivitydatabase (accessed on 20 February 2024). This has unlocked a greater understanding of cellular responses to vaccines. In a simian immunodeficiency virus (SIV) vaccination model of human immunodeficiency virus (HIV), toll-like receptors encapsulated in nanoparticles were shown to activate the innate immune cells, supporting the translation of similar adjuvants into the clinic [62]. Moreover, flow cytometry was leveraged to unanticipated effects of distinct adjuvants on the efficacy of a COVID-19 subunit vaccine. This work revealed that out of a panel of five adjuvants, AS03 was associated with the most robust CD4+ T-helper cell activation, correlating with an observed enhancement in neutralizing antibody titers [63]. The collective work to characterize both the efficacy and mechanism of this vaccine underpinned several clinical trials, eventually supporting the approval of Skycovione in the Republic of Korea (e.g., NCT04742738, NCT04750343, and NCT05007951). The ability to bridge assays and mechanisms between preclinical species and clinical settings further enables flow cytometric approaches as a key tool for understanding the cellular response to vaccines [64,65], which is critical to gaining a systems-level understanding of this therapeutic class [66]. 

There is broad translational importance to understanding mechanisms of therapeutic action on diverse cellular populations, and flow cytometry represents a key enabling tool to further this understanding. Single-cell RNA sequencing (scRNAseq) has also emerged as a technology to study transcriptomics changes with a single-cell resolution and promises to further revolutionize the drug discovery field [67]. scRNAseq and flow cytometry are highly complementary to one another, enabling direct correlation and comparison between transcriptional and protein expression changes. There are increasing examples where scRNAseq and flow cytometric datasets form an integrated systems evaluation of translational models. Zhang et al. used both techniques to understand the mechanism of action of myeloid targeting therapies in colon cancer, revealing that the presence of distinct myeloid sub-populations measured at baseline can be predictive of response to anti-CSF1R blockade or anti-CD40 agonism [68]. Integrated scRNAseq and flow cytometry approaches were used to characterize composition changes and immune-activation in a preclinical mouse model of nonischemic pressure-overload, which may explain clinical associations between heart failure and treatment responses in several settings [69]. In another example, scRNAseq and flow cytometry were leveraged to characterize immune phenotypes in a preclinical model of asthma exacerbation after corticosteroid therapy, identifying a population of IL-13 producing CD8+ memory T cells, ILC2, and basophils as key steroid-resistant populations that could drive pathogenesis [70]. Despite these studies exemplifying very different therapeutic areas and translational questions, some commonalities can be seen in the methodological integration of scRNAseq and flow cytometry. Namely, scRNAseq is emerging as a preferred technique for initial hypothesis generation, whereas flow cytometry has advantages in confirming biomarker expression at a protein level and in better quantifying mechanistic outcomes. Moreover, there are many non-overlapping applications of both scRNAseq and flow cytometry. For example, scRNAseq was leveraged to create cell maps of airway cells from Cystic Fibrosis patients [71], where flow cytometry reagents are unavailable. In contrast, flow cytometric approaches are uniquely suited to the measurement of active signaling events [72].

Collectively, flow cytometry is increasingly applied to translational research questions, in particular, to inform on cellular immune PD. The application of flow cytometry can maximize mechanistic information gained from higher species and clinical studies, playing an ethical role in minimizing the number of animals or patients required to address therapeutic questions.

## 4. Quantitative Pharmacokinetic/Pharmacodynamic Evaluation

The PK/PD relationship following therapeutic administration goes beyond qualitative association and is highly quantitative in nature. PK/PD can be closely linked in time, such as where freely diffusible small molecule drugs can have a near-instantaneous effect on target engagement and proximal signaling events in cells. The therapeutic mechanism of action may also depend on physiological effects playing out over days or weeks, particularly when a mechanism impacts changes at a cellular, tissue, or whole organism level. 

Flow cytometry is a highly relevant technique for monitoring complex cellular dynamics over time. For example, flow cytometry enabled quantitative measurement of developing thymic populations over a 10-day period, and data were used to fit an ordinary differential equation model that inferred population input, output, and death rates. The study showed that different death rates of T cells developing to the CD4 T-helper and CD8 T cytotoxic lineages led to a skew in the CD4:CD8 ratio in the periphery [73]. Combining quantitative multiparametric flow cytometry data with modeling approaches has been further applied to provide insights into cellular dynamics of activated and memory T cell homeostasis [74,75], as well as B cell activation [76].

A longitudinal tracking approach was taken to explore the pharmacology of therapeutic antisense oligonucleotides targeting the transcription factor FOXP3. The FOXP3 transcription factor is a master regulator of immune tolerance through its lineage-defining role in immunosuppressive regulatory T cells (Tregs) [77,78]. FOXP3 is a particularly attractive target in cancer, as the mechanism is overexpressed in solid tumors where it might promote immune escape and resistance to immuno-oncology drugs [79]. However, targeting this axis potentially disrupts central tolerance mechanisms and could confer an auto-immune safety risk [80,81]. Mice were dosed continuously with murine surrogate Foxp3 ASOs and showed a progressive loss of circulating FOXP3+ T cells with a concomitant expansion of activated/effector phenotype T cells. Monitoring immune populations over time showed that steady-state population stabilization was not reached for several weeks. This finding informed the optimal timepoint for exploratory safety studies, which showed no histopathological auto-immune findings, supporting a rationale for a therapeutic window [17]. These insights were also relevant to clinical biomarker monitoring, informing optimal treatment timepoints to achieve maximal dynamic effects.

With the emergence of cellular therapies such as CAR-T and TIL therapies as treatments for hematological malignancies or, more recently, melanoma [82], kinetic monitoring by flow cytometry is also relevant for monitoring drug PK. Tisagenlecleucel (Kymriah™) is an autologous CAR-T therapy approved for the treatment of relapsed/refractory large B cell lymphoma that targets CD19-expressing lymphoma cells. Flow cytometry was used alongside quantitative PCR approaches to monitor Tisagenlecleucel PK over the months following therapeutic administration [83]. CAR-T cell PK was typified by an initial reduction in circulating frequencies (redistribution), followed by an antigen-dependent expansion phase and a long persistence phase. Expansion and persistence of CAR-T cells are closely associated with clinical response over a wide dose range [84,85], a finding that extends to other CAR-T therapies [86,87,88]. The authors further suggested that despite the broad concordance of qPCR and flow cytometry methods to track CAR-T cell PK, only flow cytometry could give insights into the functionality of the CAR-T cells through phenotypic monitoring [85]. In chronic B-lymphocytic leukemia, the magnitude of CAR-T expansion following infusion could be linked back to initial donor cell memory phenotypes of autologous cells prior to manufacturing, which reflects intrinsic differences in donor cell potency [89]. Similarly, the quality and properties of TIL therapy products are also associated with efficacy. A flow cytometric assay was leveraged to measure the reactivity of TIL products to melanoma antigenic peptides bound in fluorescent MHC tetramers before and after administration. Importantly, T cell reactivity post-therapy could be almost entirely explained by the initial reactivity observed in the drug product [90]. Moreover, longitudinal monitoring of TIL therapy in melanoma patients with antigen peptide-MHC (pMHC) multimers revealed increased frequency (up to 750-fold) and long-term (>6 months) persistence of antigen-specific T cells in cohorts of responding patients [91]. Thus, whilst the flow cytometric PK assay varied (direct detection of CAR-T idiotype versus detection of TIL pMHC tetramer/multimer binding), the commonalities linking drug product and PK/PD/efficacy relationships over time were clearly defined. Insights gained from PK evaluation and phenotypic monitoring of cellular immunotherapy drugs may inform the design of next-generation approaches aimed to increase or preserve memory cell fractions [92], supporting efforts to further improve potency or extend the therapeutic benefit to solid tumors [93].

The immune system is broadly important across many diseases, and it is unsurprising that flow cytometric PK/PD evaluation has been increasingly leveraged across a spectrum of diseases and therapeutic areas. Beyond examples in oncology, Bakker et al. monitored peripheral immune cell changes in patients with atopic dermatitis following treatment with the anti-IL-4 receptor mAb therapy Dupilumab. Whilst target engagement could be measured in broad populations of immune cells 2 h after administration, the downstream PD changes were restricted to skin-homing populations of Th2-polarized immune cells and manifested over several weeks, supporting the long-term specificity of this treatment [94]. Changes in cellular composition and phenotype are also relevant in vaccine research, where they may inform the kinetic interplay between innate and adaptive immunity. For example, longitudinal measurement of cellular PD may have relevance for identifying correlates of protection beyond typically monitored antibody titers [95]. Finally, there are emerging examples where flow and related mass cytometry techniques are being used to monitor immune cell reconstitution following hematopoietic stem cell transplant [96], which have relevance for an emerging class of CRISPR-based therapies in genetic disease.

We highlight a broadening use of flow cytometry to monitor and evaluate multiparametric PK/PD relationships at a cellular and molecular level. These studies ultimately aid the understanding of therapeutic windows and optimal dosing schedules and provide insights into PD changes corresponding to therapeutic response. Nevertheless, limitations exist regarding the complexities of flow cytometric workflows and cost. Moreover, peripheral monitoring of immune populations is often indirect relative to the site-of-action of a drug in diseased tissue. As technical hurdles and cost-related limitations continue to be reduced, we anticipate expanded implementation of flow cytometry to monitor and explore PK/PD relationships.

## 5. Limitations and Opportunities for Flow Cytometric Techniques to Inform Early Clinical Decision-Making

As new drug candidates progress into the clinic, flow cytometry can be a highly relevant platform to address early clinical questions and enable decision-making on the progression or termination of a program. Early phase 1 clinical studies must evaluate the PK and safety of a molecule before dose escalating to identify the optimal phase 2 dose. In addition, it is important to generate a dataset supporting clinical proof-of-mechanism (POM), defined as evidence that a molecule reaches the desired target tissue and modulates its target and downstream biology in a manner and magnitude consistent with the therapeutic hypothesis [97]. Phase 2 trials subsequently support a clinical proof-of-concept (POC), representing early evidence of clinical efficacy to enable decision-making on progression to larger, registrational phase 3 trials [98]. POM and POC are key de-risking steps in the development path of a new medicine, and each of these milestones is associated with an increased probability of future clinical success [99]. A multi-disciplinary approach is often taken to support POM and POC milestones, including direct and/or indirect measurements of PK, PD, clinical endpoints, and mathematical modeling. The use of appropriate PD biomarkers in early-phase clinical trials has been consistently associated with higher success rates [100].

Flow cytometric techniques have been increasingly deployed in early-phase clinical trials but are most often limited to measuring exploratory endpoints in support of POM or in providing mechanistic insights enabling reverse translation. Several studies have leveraged flow cytometry or related methods such as CyTOF to identify biomarkers of response to immune-checkpoint blockade [101,102] or to monitor safety-related safety complications [101]. Clinical trials in asthma patients often rely on indirect measurements of circulating cytokines or clinical endpoints related to lung function. Fricker et al. developed a flow cytometry-based assay to measure the cellular composition of sputum, identifying mast cell prevalence as a potentially useful biomarker correlating with clinical outcomes (ref. [103]). In another example, Gonzalez-Vido et al. identified that the presence of circulating CD8+ A_4_B_7_ memory T cells was a correlate of response to the anti-A_4_B_7_ monoclonal antibody Vedolizumab [104]. One area where flow cytometry has been particularly impactful in direct clinical decision-making has been to measure receptor occupancy/target engagement of biologics [105]. Topalian et al. utilized flow cytometry to measure receptor occupancy as part of the early development program of Nivolumab (BMS-936558) [106], and a previous example in this review highlights a similar approach to measuring receptor occupancy in the development of the anti-IL-4Ra antibody Dupilumab [94].

What underlies the discrepancy between the broad use of flow cytometry in a preclinical setting versus more limited application in the clinic? One contributing factor is the complexity involved in panel design, qualification, and validation required for clinical decision-making [107,108,109,110]. Operationally, clinical trials often take place across multiple sites, leading to discrepancies in sample collection or preparation methods [111]. Efforts have been made to overcome this challenge through the introduction of best-practices for standardization. For example, the EuroFlow consortium has developed standard operating procedures and optimized staining panels to improve the diagnosis and classification of hematopoietic malignancies, available at https://euroflow.org/ (accessed on 20 February 2024) [112]. Work from the Nolan lab contributes to the understanding of best practices for phospho-flow staining, including characterization of optimal fixation and permeabilization reagents [113]. Attempts have also been made to define optimal sample shipping conditions [114,115]. Nevertheless, these recommendations are not universally implemented, require standardization of reagents and equipment that is not always feasible, and may require considerable effort to re-validate when novel biomarkers or staining panels are being measured. In addition, accessibility of sufficient samples of diseased tissues, reagent cost, instrument time, and operator skill level all contribute to higher costs for clinical flow cytometry versus other bioanalytical techniques.

We anticipate that the maturation of several technologies and analysis approaches will contribute to the evolution of clinical flow cytometry, enabling an increased clinical impact in the coming years. Firstly, the maturation of flow cytometric hardware, including mass cytometry and spectral flow cytometry, will greatly enable flexibility in panel design and expand the number of parameters that can be evaluated in a single sample [116]. Secondly, the evolution of multi-omic and AI/ML-driven data analysis approaches will enable automated, multiparametric evaluation of cell populations [117,118], which may ultimately serve to reduce the variability conferred by manual definition of cell populations and gating. Finally, platforms that can integrate cellular flow cytometric data with systems-level multi-omics datasets will further maximize insights that can be gained from clinical flow cytometry datasets [119,120]. Collectively, technology evolution is still needed to expedite a greater integration of flow cytometry into early clinical trials.

## 6. Conclusions

The convergent maturation of flow cytometry hardware, assays, and data analysis is having a fundamental impact on drug discovery and translation research. The continuous evolution of flow cytometry methods provides advanced alternatives to previous gold-standard techniques and enables new insights into pharmacology, translational models, and patient samples. We anticipate that further integration of spatial and fluorescence imaging, reagent availability, and throughput will see flow cytometry continue to evolve as a pivotal technique. Continued standardization will also be important to broaden the application to less specialized operators. Importantly, flow cytometry bridges molecular and cellular biology readouts in the same assay, which is critically important to understand and enable the discovery and early development of many emerging classes of therapeutics. In the coming years, we envisage an accelerated role for flow cytometric techniques to advance preclinical pipelines in the biopharma industry and a further emergence as a tool to enable clinical decision-making.

## Figures and Tables

**Figure 1 ijms-25-03851-f001:**
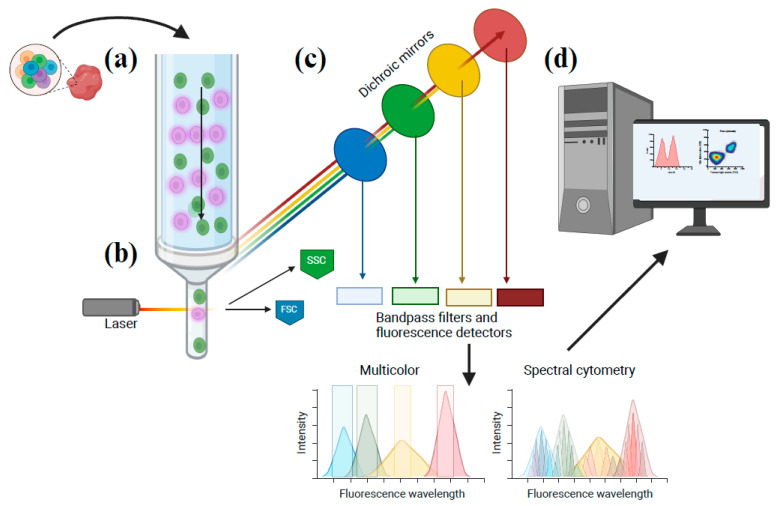
Advances in flow cytometry workflow. (**a**) Single-cell suspensions are prepared and stained with fluorescent antibodies and dyes. The sample is pressurized and focused so that cells pass into a flow cell in ‘single-file’ for interrogation by a series of light-emitting lasers. (**b**) Lasers excite fluorescent dyes and fluorophores, and emitted light travels through fiber optic cables. (**c**) In conventional multicolor flow cytometry, light of specific wavelengths is directed toward photon detectors through a series of dichroic mirrors and bandpass filters. In a recent advance, spectral flow cytometry splits photons by wavelength using a prism or grating to be collected across a spectrum of detectors, and individual fluorophore signals are deconvoluted computationally with a spectral unmixing algorithm. (**d**) Flow cytometry software is utilized to visualize multiparametric data with single-cell resolution. Advanced analysis and visualization software enables approaches including automated gating, cell clustering, cell population identification, and omics integration.

**Figure 2 ijms-25-03851-f002:**
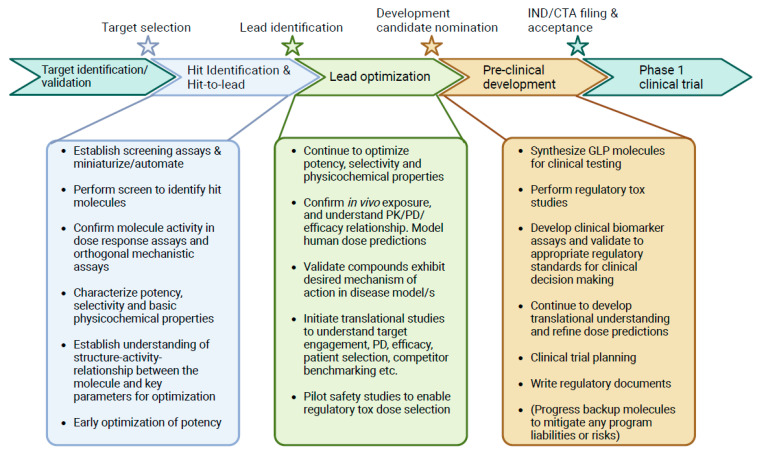
Generalized drug discovery pipeline and activities. Biopharmaceutical drug discovery programs follow a step-wise progression, shown inside chevrons. Stars highlight key milestones in a program that can gate the next waves of activities once achieved. Boxes highlight some key activities performed during three stages of drug discovery. Bulleted examples are typical of small molecule discovery; activities may vary for other modalities. GLP—good laboratory practices (regulatory standard); IND—investigational new drug (FDA regulatory documentation for Ph1 trials); CTA—clinical trial application (European Medicines Agency regulatory documentation for Ph1 trials).

**Figure 3 ijms-25-03851-f003:**
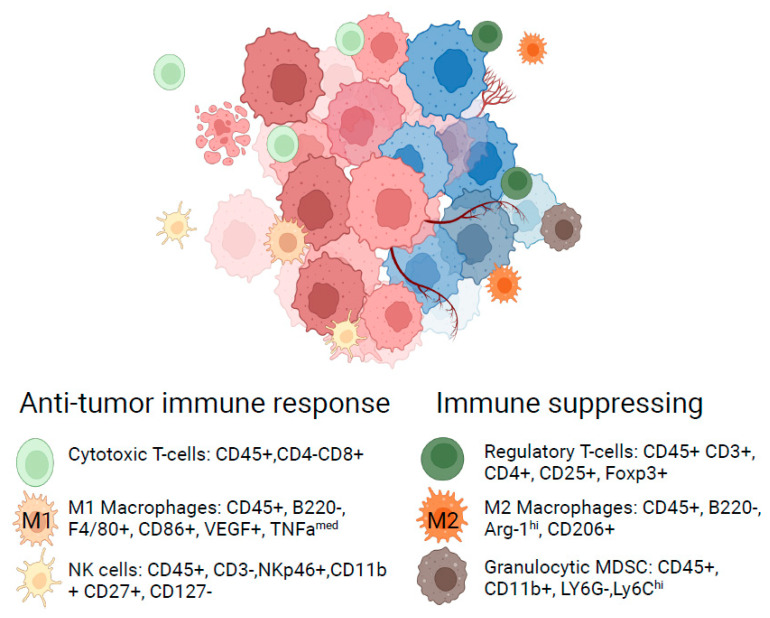
Phenotype of immune cell populations from mouse syngeneic tumors by flow cytometry. Biomarker expression examples depict common detection strategies for immune cells isolated from mouse syngeneic tumors. The hematopoietic cell lineage is identified by surface expression of CD45. Populations of immune cells can be further subsetted through broader measurement of cellular antigen expression. Flow cytometry enables the identification of cells associated with an anti-tumor cytotoxic immune response or immune-suppressing cell populations that hold anti-tumor immunity in check. Immuno-oncology therapies aim to potentiate cytotoxic immunity or mitigate immunosuppressive cells or mechanisms in the tumor microenvironment, which can lead to changes in immune cell composition or phenotype.

**Table 1 ijms-25-03851-t001:** Key pharmaceutical properties that are optimized during the drug discovery process.

Therapeutic Property	Examples of Parameters Considered	Objective
Potency	Biochemical potency (intracellular targets) Affinity/avidity Cellular potency Potency and activity in primary cells	Early screening hits typically lack sufficient potency for clinical activity at a feasible dose and require iterative optimization to improve potency.
Safety and selectivity	Activity vs. homologous proteins and isoforms Selectivity vs. unrelated proteins Selectivity vs. targets associated with adverse drug reactions	Optimize primary target specificity while minimizing secondary interactions conferring safety risks.
Pharmacokinetics and drug exposure	Solubility and permeability Transporter activity to optimize drug exposure and/or brain penetrance Metabolic/Plasma stability Immunogenicity In vivo pharmacokinetic properties (e.g., bioavailability, Cmax, Ctrough, AUC, half-life)	Optimization of properties that affect drug exposure in target tissues. These typically include characteristics related to absorption, distribution, metabolism, and excretion (ADME) of a molecule.
Therapeutic functionality	Active transport Endocytosis or internalization Degradation Payload release Cellular or molecular complex formation and stability	Properties that affect the molecular mechanism of action.

**Table 2 ijms-25-03851-t002:** Selected examples of flow cytometry assays used to drive hit identification and lead optimization in drug discovery.

Lead Author	Disease Indication	Target	Modality	Use of Flow Cytometry
Tuijnenburg [7]	Auto-immunity	MTOR pathway (via phenotypic screen)	Small molecule	Phenotypic screen for regulators of auto-antibody production
Schardt [11]	Virology	SARS-CoV-2 neutralizing antibodies	Antibody	Sorting of memory-specific B cell clones for expansion/antibody production
Zhou [13]	Oncology	EGFR	Antibody	Characterization and rank ordering of antibodies based on binding affinity
Revenko [17]	Oncology	FOXP3	ASOs	Potency ranking of ASOs in primary cells
McDermott [21]	Oncology	CLDN6	Antibody/ADC	Characterize selectivity versus related family members
Nilsson [25]	Oncology	JAK1	Small molecule	Characterize selectivity (JAK1 vs. JAK2)
Kirkland [28]	Virology	Shiga toxin	Antibody	Evaluate immune cell activation
Mandrup [32]	Immuno-Oncology	N/A	Bi-specific antibody	Evaluate the effect of half-life modifications on efficacy
Parameswaran [34]	Oncology	CD6	ADC	Measure internalization efficiency
Li [41]	Oncology	HER2	ADC	ADC payload release versus target and bystander cells

## Data Availability

No new data were created or analyzed in this study. Data sharing is not applicable.

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
