# Peer review of "Applications of Flow Cytometry in Drug Discovery and Translational Research"

_ijms, 2024, doi:10.3390/ijms25073851_

Round 1

Reviewer 1 Report

Comments and Suggestions for Authors

In this manuscript, Ullas and Sinclair reviewed the application of flow cytometry in drug discovery and translational research. The topic is relevant and timely. The manuscript is well structured and contains information that is of interest to a broad range of readers. I have the following comments.

Major points

1. The authors selected examples where flow cytometry was employed to facilitate drug discovery under section 2 ‘Hit identification and lead optimization’. It is advised to exhibit this information in a table, where key information of the selected studies is summarized, e.g. disease indication, target, modality, utility of flow cytometry and references, etc. It would also be good to know the development stage of these molecules.

2. The authors could consider discussing the utility of flow cytometry in revealing unknown functions of known drugs. For example, some cancer targeting drugs may have previously unknown immune modulatory functions. Another example is the use of flow cytometry in understanding effects of vaccines.

3. The authors discussed the application of flow cytometry in CAR-T therapy. This discussion should include addition examples for TIL therapy.

4. scRNAseq has been widely used to profile changes in pre-clinical models and patients. The authors should discuss and exemplify the role of flow cytometric analysis to scRNAseq.

5. Under ‘limitations’, the author could give recommendations on validation of reagents or protocols for flow cytometric analysis.

Minor points

1. There are some inconsistencies in the spelling, e.g. PD-L1 or PDL-1 or PD(L)-1.

2. ‘With the emergence of cellular therapies such as CAR-T and TIL therapies as treatments for hematological malignancies,..’. TILs are isolated from solid tumors and the therapy is approved to treat melanoma.

Author Response

We would like to thank both reviewers for their positive feedback and helpful suggestions to improve our review article. The suggested improvements have been incorporated into our resubmitted manuscript, including several additional sections, 19 additional references and amendment of minor typographical errors. We feel we’ve addressed the feedback from both reviewers with these additions, and welcome any further comments or guidance from the editors.

Reviewer 1:

Major points:

  1. The authors selected examples where flow cytometry was employed to facilitate drug discovery under section 2 ‘Hit identification and lead optimization’. It is advised to exhibit this information in a table, where key information of the selected studies is summarized, e.g. disease indication, target, modality, utility of flow cytometry and references, etc. It would also be good to know the development stage of these molecules.

We’ve incorporated the suggestion to include a summary table of the studies from the hit identification and lead optimization section for select papers from this section (pg. 7).

We began to also collate information on the development stage of the molecules, however this was challenging. Novel therapeutics are discovered through a series of screening and optimization steps, it is not always clear which therapeutics can be directly attributed to published hit identification and lead optimization activities. Moreover, nomenclature for therapeutic molecules is often changed during the process of development candidate nomination, which can lead to further challenges and discrepancies when tracking back to the earlier discovery publications. Finally, given that it can take several years for a program to progress from hit-identification to the clinic there was a lack of information on some of the programs in terms of their current status and whether they represented currently active programs.

Given these challenges to directly attribute clinical drug candidates to early discovery efforts, we were not able to include the ‘development stage’.

  1. The authors could consider discussing the utility of flow cytometry in revealing unknown functions of known drugs. For example, some cancer targeting drugs may have previously unknown immune modulatory functions. Another example is the use of flow cytometry in understanding effects of vaccines.

This is an important consideration, and whilst we have touched on some examples in our manuscript, the key points may have not been emphasized explicitly. To keep the structure of our manuscript centered around the biopharmaceutical paradigm of drug discovery (rather than drug repurposing efforts), we proposed to include additional discussion and references in two sections of our revised manuscript.

Firstly, we have added additional context and a key reference on pg. 9, highlighting the impact of flow cytometry on revealing unexpected immune-modulatory functions of tumor-targeted agents. The Sceneay et al. review highlights a further ~10 examples in greater detail.

We also added another example for the use of flow cytometry in understanding vaccines on pg. 10. Arunachalam leveraged multi-omics and flow cytometry to provide a highly impactful mechanistic rationale for adjuvant selection of a COVID-19 antibody, work that supported the approval of Skycovione in the Republic of Korea, and we included some additional discussion around the role of flow cytometry in enabling a complete understanding of ‘system vaccinology’ through measurement of cellular immune responses.

  1. The authors discussed the application of flow cytometry in CAR-T therapy. This discussion should include addition examples for TIL therapy.

We have included some additional discussion on the application of flow cytometry in TIL therapy on pg. 12. We thank Reviewer 1 for this suggestion.

  1. scRNAseq has been widely used to profile changes in pre-clinical models and patients. The authors should discuss and exemplify the role of flow cytometric analysis to scRNAseq.

We similarly thank Reviewer 1 for this interesting suggestion to compare and contrast scRNAseq and flow cytometry. We have included an additional paragraph on pg. 10, highlighting both the complementarity of the techniques in addition to more unique applications that answer distinct translational questions.

  1. Under ‘limitations’, the author could give recommendations on validation of reagents or protocols for flow cytometric analysis.

We acknowledge the opportunity to improve our review with the addition of some reference to validation/standardization approaches for flow cytometry, as this was also highlighted by reviewer 2. We have included some additional discussion points and considerations on pg. 13.

Minor points

We amended the minor errors highlighted by reviewer 1.

Reviewer 2 Report

Comments and Suggestions for Authors

The review by Ullas and Sinclair describes the importance and contributions of flow cytometry technology to the drug discovery and clinical trials efforts.

This work is of increased interest for all those who are seeking to understand the variety of applications of flow cytometry through the lens of biomedical discoveries.

The review thoroughly addresses all steps of drug discovery and describes important studies where flow cytometry has made a difference towards development of translatable therapeutic molecules. This makes the work interesting also for experts in flow cytometry that are not very familiar with the drug discovery and development process.

The reading is very enjoyable – authors use clear English and a rich vocabulary; sentences flow naturally.

However, minor errors have been identified that should be corrected, before publication. Also, there are a few omissions that should be corrected. These are a few suggestions:

-page 1, line 24 – “As a result,flow cytometry…” instead of “As a result flow cytometry…”

-page 1, line 26 – “termed polychromatic or multicolourflow cytometry”  instead of “termed polychromatic flow cytometry”

-page 1, line 32 – add parentheses to figure citation here and throughout text

-page 1, line 39 – “measure conjugated antibody by mass of attached metal isotope” instead of “measure conjugated antibody by mass”

-page 1, line 42 – “ fluorescent antibody or dyelocalization” insead of “ fluorescent antibody localization”

-page 3, line 95 – “modulators of macrophage…” instead of “modulators macrophage…”

-page 7, line 256 – “further insight intocellular cross-talk” instead of “further insight cellular cross-talk”

-page 8, line 277 – “small moleculetargetingthe …” instead of “small molecule targetedthe …”

-page 11, line 447 – “in supportofPOM” insead of “in support POM”

Also, authors raise the issue of differences in sample collection and processing between centers in multi-center clinical trials (page 12, lines 466-467, ref 97). The efforts made by EuroFlow consortium for standardization in flow cytometry as well as work from Gary Nolan in establishing good practice and protocols in flow cytometry (especially PhosphoFlow) analyses as well as sample preparation for shipping at large distances should be mentioned.

Given the above observations, I propose the acceptance of the manuscript after minor revision.

Author Response

The minor errors noted by reviewer 2 have been corrected. In addition, we incorporated the suggestion to discuss the development of SOPs for clinical flow cytometry and include key references (pg. 13), which was also highlighted by Reviewer 1.

We would like to thank reviewer 2 for the positive comments on our paper, and helpful suggestion that strengthens the clinical flow cytometry section.

Round 2

Reviewer 1 Report

Comments and Suggestions for Authors

The revision has addressed all my questions.